# Structural variation underlies functional diversity at methyl salicylate loci in tomato

**Manoj Sapkota[1], Lara Pereira[1], Yanbing Wang[1], Lei Zhang[1], Yasin Topcu[1], Denise Tieman[2], Esther van der Knaap[1] \***

1 Institute of Plant Breeding, Genetics, and Genomics, University of Georgia, Athens, Georgia, United States of America, 2 Horticultural Sciences, University of Florida, Gainesville, Florida, United States of America

\* esthervanderknaap@uga.edu

**Data Availability Statement:** All the raw sequencing data used in this study are publicly available in NCBI (https://www.ncbi.nlm.nih.gov/;

## Abstract

Methyl salicylate is an important inter- and intra-plant signaling molecule, but is deemed undesirable by humans when it accumulates to high levels in ripe fruits. Balancing the trade-off between consumer satisfaction and overall plant health is challenging as the mechanisms regulating volatile levels have not yet been fully elucidated. In this study, we investigated the accumulation of methyl salicylate in ripe fruits of tomatoes that belong to the red-fruited clade. We determine the genetic diversity and the interaction of four known loci controlling methyl salicylate levels in ripe fruits. In addition to *Non-Smoky Glucosyl Transferase 1* (*NSGT1*), we uncovered extensive genome structural variation (SV) at the *Methylesterase* (*MES)* locus. This locus contains four tandemly duplicated *Methylesterase* genes and genome sequence investigations at the locus identified nine distinct haplotypes. Based on gene expression and results from biparental crosses, functional and non-functional haplotypes for *MES* were identified. The combination of the non-functional *MES* haplotype 2 and the non-functional *NSGT1* haplotype IV or V in a GWAS panel showed high methyl salicylate levels in ripe fruits, particularly in accessions from Ecuador, demonstrating a strong interaction between these two loci and suggesting an ecological advantage. The genetic variation at the other two known loci, *Salicylic Acid Methyl Transferase 1* (*SAMT1*) and tomato *UDP Glycosyl Transferase 5* (*SlUGT5*), did not explain volatile variation in the red-fruited tomato germplasm, suggesting a minor role in methyl salicylate production in red-fruited tomato. Lastly, we found that most heirloom and modern tomato accessions carried a functional *MES* and a non-functional *NSGT1* haplotype, ensuring acceptable levels of methyl salicylate in fruits. Yet, future selection of the functional *NSGT1* allele could potentially improve flavor in the modern germplasm.

## Author summary

Tomato fruit release a wide variety of volatiles and these volatiles are responsible for its unique aroma. One of the important volatiles in tomato flavor is methyl salicylate. Methyl salicylate is important for plant defense, but tomato taste panels routinely associate this volatile with low liking as it features the characteristic odor of wintergreen. Several

SRA: SRP150040, SRP045767, SRP094624, and PRJNA353161).

**Funding:** This work was funded by NSF IOS 2151032 (EvdK and DT) and NSF IOS 1732253 (EvdK). M.S. is the recipient of the John Ingle Innovation in Plant Breeding Award and Roger Boerma Plant Breeding Excellence Scholarship Award. The funders had no role in study design, data collection and analysis, decision to publish, or preparation of the manuscript.

**Competing interests:** The authors have declared that no competing interests exist.

enzymes involved in the biosynthesis of methyl salicylate have been identified in tomato but the genetic diversity at the gene region has not yet been fully explored. In this study, we investigate the genetic variations at the four known gene regions controlling fruit methyl salicylate levels using a diverse collection ranging from red-fruited wild tomatoes to modern and heirloom varieties. We identified extensive genomic structural variations at two of the four loci. The genetic interaction between two loci determines the volatile levels. We also showed dramatic selection pressures ensued at these loci during evolution and domestication of the tomato. Our study greatly improves the understanding of the genetic basis of methyl salicylate production in tomato fruits and gives breeders far better tools to create agrosystem-specific adapted commercial germplasm.

## Introduction

Plants synthesize and emit many volatile organic compounds that play key roles in reproduction, communication, and stress responses. The volatiles emitted from flowers allow plants to attract and guide pollinators to ensure fertilization and reproduction [1,2], whereas aroma from fruits allure frugivores to assist in seed dispersal [3–5] and fend off pre- and post-harvest diseases [5–7]. Volatiles play important roles in intra- and inter-plant communications as these compounds facilitate the signaling about the danger to distant parts of same or to neighboring plants [8,9]. Furthermore, volatiles can act as direct defenses, either by repelling the danger [10,11], or intoxicating herbivores and pathogens [12,13]. Volatiles may also serve as signals to attract herbivores' natural enemies such as parasitoids and predators [14–18] and even predatory birds [19].

The complex tomato fruit volatile aroma is produced by derivatives from six major biochemical pathways: (i) fatty acid or lipid-derived; (ii) branched chain amino acid (BCAA)-derived such as leucine (Leu) and isoleucine (Ile); (iii) phenylalanine (Phe)-derived including phenylpropanoids and phenolics; (iv) terpenoid-derived; (v) carotenoid-derived; and the (vi) acetate ester volatiles [20–22]. Tomato fruit release a wide variety of volatiles, with emission rates peaking at ripening [23]. These volatiles comprise only $10^{-7}$ to $10^{-4}$ of the fresh fruit weight and are responsible for its unique aroma [24]. In general, well-liked tomatoes feature a complex volatile aroma that is high in certain volatiles and low in others, and properly balanced in yet other volatiles [20,25].

Red-fruited tomato is edible and evolved in the tomato clade of the *Solanum* genus in the Solanaceae family. The red-fruited *S. pimpinellifolium* (SP) last shared a common ancestor with the green-fruited species approximately two million years ago [26,27]. Within the red-fruited clade, SP evolved into wild *S. lycopersicum cerasiforme* (SLC) that then became domesticated into *S. lycopersicum lycopersicum* (SLL) around seven thousand years ago [28,29]. The huge variation in volatile production in tomato from SP to SLL indicates that its domestication placed strong selection pressures on this trait [25]. One of the important volatiles in tomato flavor is methyl salicylate which is produced in the Phe-derived pathway. Tomato taste panels routinely associate this volatile with low liking as it features the characteristic odor of wintergreen. However, some varieties with moderate to high levels of methyl salicylate rate well for overall liking. This may be the result of concentrations of or interactions with other volatiles resulting in overall good flavor [25,30,31].

Several enzymes involved in the biosynthesis of methyl salicylate and its glucoside conjugation have been identified in tomato (Fig 1) [32–35]. Methyl salicylate is synthesized from salicylic acid, which is produced from the isochorismate and the Phe-derived pathways [36]. The

Isochorismate pathway    Phenylalanine pathway

Salicylic acid

*SAMT1*    *MES1-4*

Methyl salicylate

*UGT5*    *Glycosidases*

disaccharide

Diglycoside conjugate

*NSGT1*

trisaccharide

Triglycoside conjugate
non-convertible to free volatile

**Fig 1. The biosynthesis and glucoside conjugation pathway of methyl salicylate in tomato.** The gray boxes represent the two pathways leading to salicylic acid. The arrows represent the direction of the reaction. The enzyme catalyzing the reaction is listed next to the arrow. The genes names are: *SAMT1*: *Salicylic Acid Methyltransferase 1*; *MES1-4*: *Methylesterase 1–4*; *UGT5*: *UDP-glycosyltransferase 5*; *NSGT1*: *Non-smoky Glycosyltransferase 1* [32–35]

process of methylation of salicylic acid into methyl salicylate is catalyzed by the enzyme SALI-CYLIC ACID METHYLTRANSFERASE 1 (*SlSAMT1; Solyc09g091550*) [32]. Another group of enzymes, METHYLESTERASE 1–4 (*SlMES1-4*: *Solyc02g065240-80*), demethylates methyl salicylate to salicylic acid to maintain homeostasis [35]. The methyl salicylate can also be glycosylated into glycoconjugate compounds for transport and storage in the vacuole [37]. The two major enzyme families involved in the formation and degradation of glycosylated methyl salicylate are: glycosyltransferases and glycosidases. Glycosyltransferases (eg. *SlUGT5; Solyc01g095620)* catalyze the formation of diglycoside conjugates [33]. These diglycoside conjugates are hydrolyzed to methyl salicylate by glycosidases. One glycosyl transferase, *Non-Smoky Glycosyltransferase 1* (*NSGT1; Solyc09g089585*), encodes a fruit ripening-induced enzyme that adds a glucose moiety to the methyl salicylate diglycoside conjugate resulting in a triglycoside. This triglycoside conjugation is irreversible and prevents the emission of the volatile upon tissue disruption and ripening [21,34]. Of the enzymes in the methyl salicylate biosynthesis and conjugation pathway, loss of function in either *SlMES1-4* or *NSGT1* results in high methyl salicylate in fruits [34,35].

*NSGT1* is known for the glycosylation of guaiacol, a smoky volatile that, similarly to methyl salicylate, negatively affects consumer liking. Based on genome structural variations (SV), *NSGT1* is represented by five haplotypes that are grouped into functional and non-functional categories based on the presence of a functional enzyme [38]. Similarly, *MES* also exhibits SVs [35], but the diversity at this locus has not yet been fully explored in tomato. The genetic diversity at the other two known genes, *SlSAMT1* and *SlUGT5*, has also not been explored in red-fruited tomato. The genetic characterization of these loci can provide evolutionary insights and lead to the identification of haplotypes that might be beneficial for crop improvement.

In this study, we analyzed a unique and genetically well-characterized tomato collection of fully wild SP, semi-domesticated SLC and ancestral SLL landraces from South and Central America (Varitome Collection) as well as a subset of cultivated accessions that includes heirloom and modern types (SLL_CUL). Collectively, they are referred to as red-fruited tomato in this study. We investigated the genetic diversity at the known genes controlling methyl salicylate levels in tomato fruits and uncovered extensive genome SV underlying the trait in the red-fruited tomato clade. We also show that different selection pressures ensued at these loci during domestication.

## Results

### Distribution of methyl salicylate levels in ripe, red-fruited tomato fruits

Quantification of fruit methyl salicylate in 166 accessions from the Varitome Collection and 143 cultivated types showed that the levels ranged from 0.003 nanogram per gram fruit weight per hour (ng/gfw/hr) to 29.360 ng/gfw/hr, representing a 9000-fold difference (Fig 2A). Most accessions showed less than 1 ng/gfw/hr of methyl salicylate in the fruits. Significantly higher levels were found in SP from Northern Ecuador and SLC from Ecuador, whereas ancestral landraces, SLL, featured low levels of the volatile. Most of the SLL_CUL produced low levels of methyl salicylate with a few exceptions. We observed variation in other sub-populations, but overall, the fruits emitted low levels of the volatile (Fig 2B).

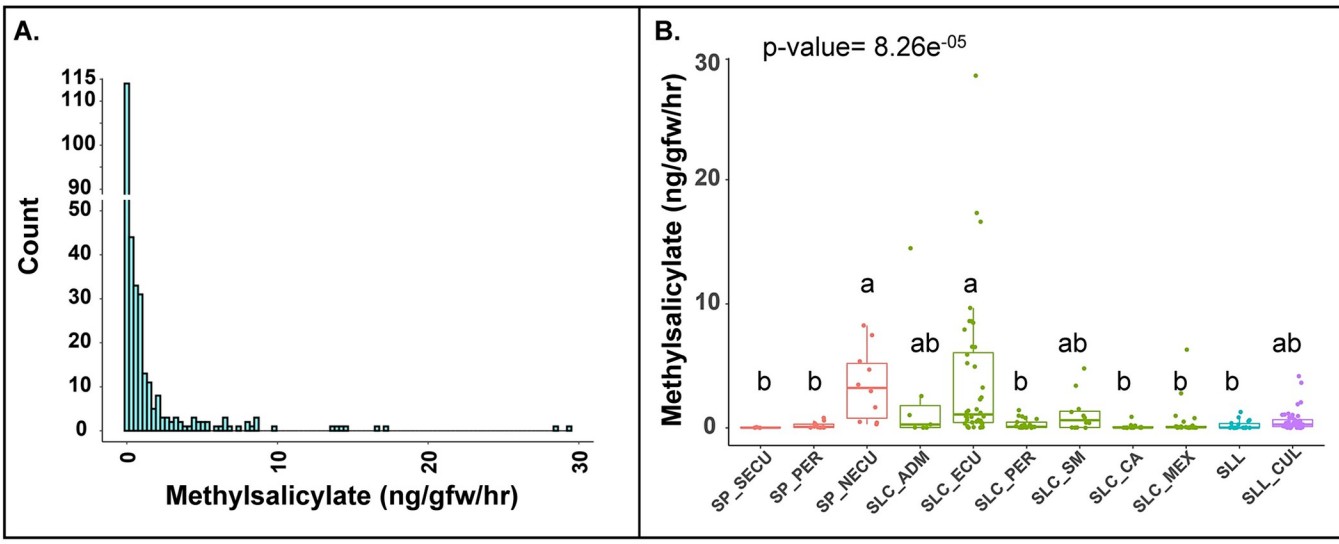

**Fig 2. Methyl salicylate levels in tomato fruits in the red-fruited tomato germplasm.** (**A**) Histogram representing the distribution of methyl salicylate in ripe tomato fruits from the Varitome Collection and heirlooms. X-axis represents the methyl salicylate levels in fruits whereas, y-axis denotes the count. (**B**) Methyl salicylate levels in tomato accessions grouped by sub population. SP_SECU: SP from Southern Ecuador, SP_PER: SP from Peru, SP_NECU: SP from Norther Ecuador, SLC_ADM: SLC admixture, SLC_ECU: SLC from Ecuador, SLC_PER: SLC from Peru, SLC_SM: SLC from San Martin, SLC_CA: SLC from Central America, SLC_MEX: SLC from Mexico, SLL: SLL in Varitome Collection and SLL_CUL: modern and heirloom accessions. The letters in the boxplots indicate the significant differences among different sub population evaluated by Duncan's test ($\alpha < 0.05$). The x-axis represents different sub population whereas, y-axis denotes the methyl salicylate levels in fruits.

## SNP, INDEL and SV-based GWAS of methyl salicylate levels in ripe tomato fruits

To investigate the genetic bases of the variation in methyl salicylate levels, we performed a genome wide association study (GWAS) using the whole genome variant information of the Varitome Collection. We identified several GWAS loci associated with fruit methyl salicylate levels (Fig 3 and S1 Table) using Single Nucleotide Polymorphisms (SNPs), Insertion-Deletions (INDELs) and Structural Variants (SVs). The identified GWAS loci included *MES* on chromosome 2 (chr 02), *NSGT1* on chr 09, and *SAMT1* also on chr 09. For *MES*, the locus was identified by all three variant types, namely a 4.7 kb deletion at position SL.40ch02:34,464,833, a SNP at SL4.0ch02:34,895,803, and an INDEL at SL4.0ch02:34,943,905 (Fig 4A–4C). The *MES* locus contains four *Methylesterase* genes (*Solyc02g065240-80*) and the 4.7 kb deletion partially spans the third exon of *Solyc02g065260* (*SlMES3*), whereas the SNP and INDEL mapped 400 kb and 467 kb downstream of the *MES* locus, respectively. The accessions carrying the 4.7 kb deletion produced significantly higher amounts of methyl salicylate compared to the accessions without the deletion (Fig 4A) and the variant explained 28% of total methyl salicylate variation in the Varitome Collection. Similar to the SV at *MES*, accessions having alternate alleles of the associated SNP and INDEL produced significantly higher levels of methyl salicylate in tomato fruits (Fig 4B and 4C). For *NSGT1*, the locus was identified by a SNP at SL4.0ch09:65,404,388 (4.3 kb downstream of *NSGT1*) and an INDEL at SL4.0ch09: 65,401,256 (7.4 kb downstream of *NSGT1*), explaining 40% of the variation in the population. These overlapping GWAS loci carried three alleles: the reference allele, the alternate (SNP or INDEL) allele, or the 15 kb deletion. Accessions with the deletion at the locus showed significantly higher levels of methyl salicylate compared to accessions without the deletion (Fig 4D and 4E). For *SAMT1*, the locus was detected by a SNP at SL4.0ch09: 67,197,164 which was located 293 kb downstream of the gene. The accessions carrying the reference allele (GG) produced lower

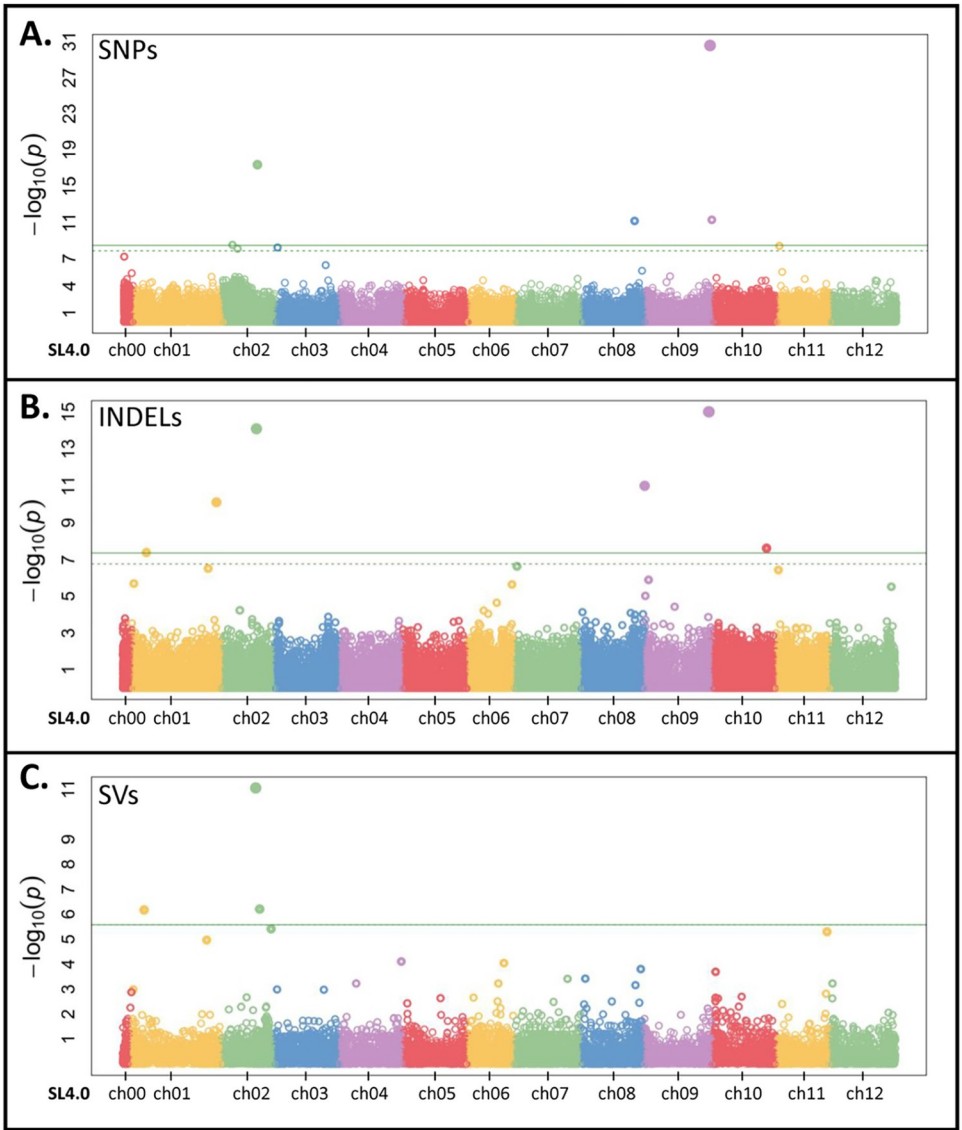

**Fig 3. Genome wide association study (GWAS) for methyl salicylate levels in the Varitome Collection.** (**A**) GWAS based on SNPs. (**B**) GWAS based on INDELs. (**C**) GWAS based on SVs. The x-axis represents different chromosomes and y-axis denotes the $-\log_{10}$(p-value) of the variants.

levels of methyl salicylate compared with the alternative allele (GA) (Fig 4F). Similar to *SAMT1*, the closest significant variant for *UGT5* on chr 01 was an INDEL (SL4.0ch01: 85,548,458) that mapped to approximately 6.5 Mb downstream the gene. For this variant, all accessions with reference allele produced low methyl salicylate levels in fruits, whereas lines with a nucleotide insertion showed significantly higher levels of the volatile (Kruskal-Wallis test p-value: 6.4e$^{-06}$) (Fig 4G).

## Genome landscape of *MES, SAMT1* and *UGT5* loci in the red-fruited tomatoes

Previously, five haplotypes of *NSGT1* were identified based on the SVs at the locus of which haplotype IV and V were associated with higher methyl salicylate levels [38]. To gain more

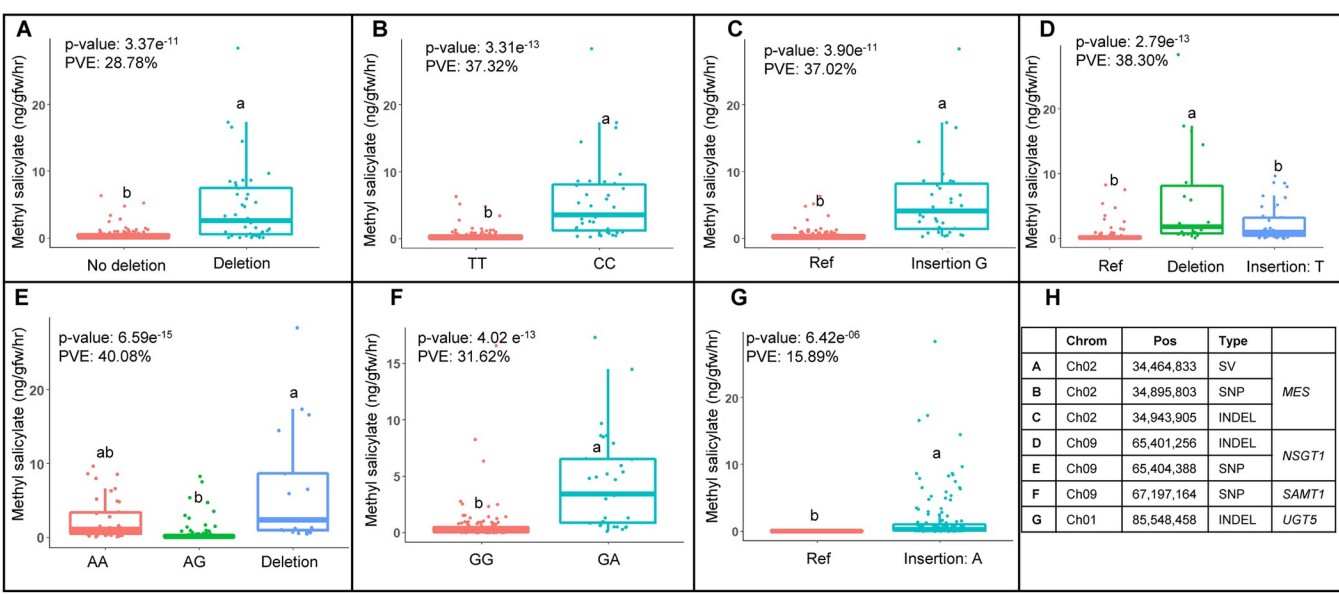

**Fig 4. Phenotypic distribution of methyl salicylate levels at the alleles of significant loci at *MES*, *NSGT1*, *SAMT1*, and *UGT5*.** Boxplots representing the distribution of methyl salicylate levels among different alleles of the (**A**) SV, (**B**) SNP, and (**C**) INDEL identified at *MES* locus from GWAS. Boxplots representing the distribution of methyl salicylate levels among different alleles of the (**D**) INDEL and (**E**) SNP identified at *NSGT1* locus from GWAS. (**F**) Boxplots representing the distribution of methyl salicylate levels among different alleles of SNP identified at *SAMT1* locus from GWAS (**G**) Boxplots representing the distribution of methyl salicylate levels among different alleles of the INDEL identified at *UGT5* locus from GWAS. (**H**) The table provides the information of significant variants identified from GWAS along with the known gene loci. The reference alleles are colored red. The p-value is obtained from Kruskal-Wallis test. The percentage of variation explained (PVE) is calculated using respective variants in the ANOVA model. The letters in the boxplots indicate the significant differences among different alleles evaluated by Duncan's test ($\alpha < 0.05$).

insight about the haplotype diversity of the other genes in this pathway, we investigated the genomic landscape at *MES*, *UGT5* and *SAMT1* loci with the focus on SVs. For *MES*, we compared the genomic sequences of the three highest methyl salicylate accumulating accessions (BGV006852: 28.388 ng/gfw/hr, BGV006931: 17.316 ng/gfw/hr and BGV006148: 14.472 ng/gfw hr) and the three lowest accumulating accessions (BGV007151: 0.0094 ng/gfw hr, BGV008098: 0.0097 ng/gfw/hr and BGV007867: 0.0099 ng/gfw/hr). The high methyl salicylate producing accessions carried several deletions at *MES* and multiple mismatches when compared to the Heinz1706 reference genome. On the other hand, the genomic landscape of the low methyl salicylate producing accessions corresponded more closely to the Heinz1706 reference genome (S1 Fig). Further comparisons based on SVs (S2 Table) showed additional haplotypes in red-fruited tomato to a total of nine (Fig 5 and S3 Table). We next investigated each of the predicted genes at *MES* to ascertain functionality with respect to the potential to encode a full-length protein (S2 Fig). The results showed the predicted functional genes at the locus ranged from zero (haplotype 9) to four (haplotype 7). *MES1* was likely functional in every haplotype except 9. *MES3* was truncated in several haplotypes, missing 97 amino acids from its C terminal end except in haplotype 1. It is known that the truncated allele of *MES3* retains at least partial enzyme activity and is therefore considered functional [35]. For *MES2* and *MES4*, several haplotypes appeared to carry a functional allele for one or both genes. *MES4* was either full length or was truncated resulting from a 48 bp deletion in the first exon (S2 Table).

We next constructed a phylogenetic tree using the genome sequence of *MES* including 10 kb that flank the four genes at the locus (Fig 6). Haplotype 1 was found only in a few accessions that did not cluster with any other haplotype (Fig 6 and S4 Table). Haplotype 3 was also distinct and differed from haplotype 1 by a transposon insertion at the C terminal end of *SlMES3*

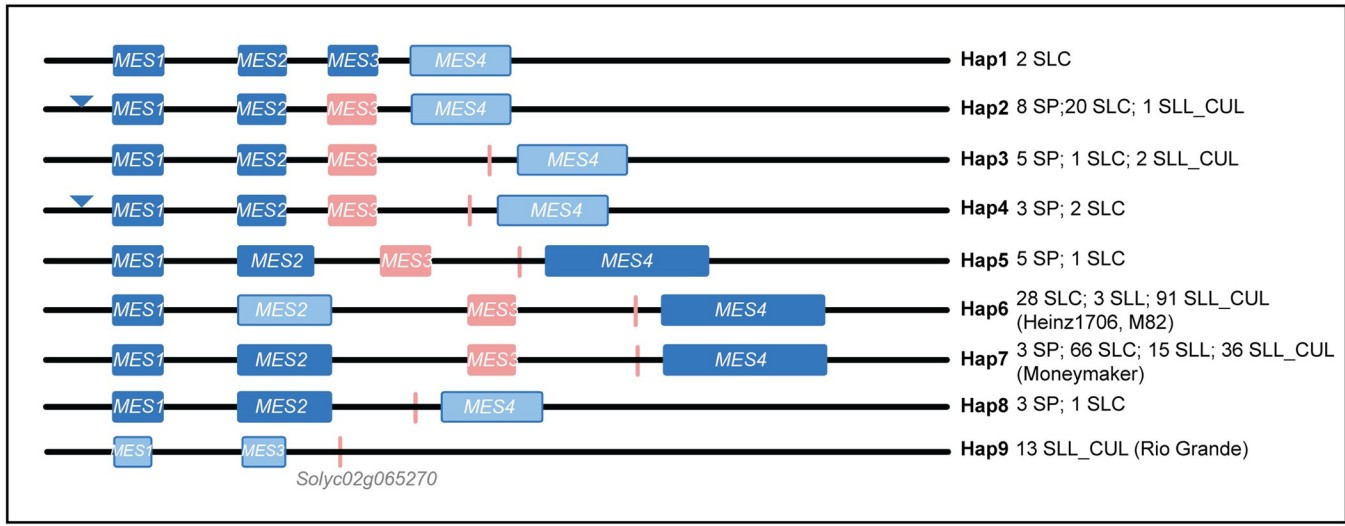

**Fig 5. Schematic representation of nine haplotypes at the *MES* locus.** Blue inverted triangles represent deletions, dark blue boxes indicate likely functional genes, light blue boxes indicate non-functional genes whereas the light red boxes represent truncated genes. Next to each haplotype are the accessions belonging to the haplotype.

resulting in a truncated version of the gene and a remaining protein segment that was annotated as *Solyc02g065270* (S3 Fig). Haplotypes 2 and 4 were also more ancestral, both showing a deletion in the promoter of *SlMES1* compared to other haplotypes. Haplotype 2 carries the 48 bp deletion in the first exon of *SlMES4*, truncating the protein to 119 amino acids instead of 279 amino acids, whereas haplotype 4 has an intergenic deletion between *SlMES3* and *SlMES4*. Haplotypes 2 and 4 were clustered together implying a close evolutionary relationship in addition to a similar SV structure. The most common haplotypes were 6 and 7, found in 122 and 120 accessions, respectively. Haplotype 6 included the reference genome Heinz1706 and M82 and haplotype 7 included Moneymaker. Compared to haplotype 5, haplotypes 6 and 7 carried transposon insertions in one of the introns of *SlMES2* and in the intergenic region between *SlMES2* and *SlMES3*. Haplotype 7 was the only haplotype for which all four genes were predicted functional, whereas haplotype 6 carried a premature stop codon in *SlMES2*, likely rendering this gene non-functional (S4 Table). Therefore, haplotype 6 appeared to be derived from haplotype 7. Haplotype 8 clustered in a separate subclade and lacked *SlMES3* as well as a non-functional *SlMES4* due to a frameshift (at SL4.0ch02:34,471,277) and three missense (at SL4.0ch02:34,471,287, SL4.0ch02:34,471,546, and SL4.0ch02:34,475,508) mutations in the latter gene. The last haplotype, haplotype 9 consisted of several large deletions at the locus resulting in truncation, internal deletions, and elimination of all four *MES* genes, and was found only in 13 SLL_CUL accessions (*e.*g., Rio Grande in S4 Fig). Clearly, the huge variation in haplotype diversity even among the fully wild accessions, suggested that selection predating domestication shaped the genome structure at this locus.

For *SAMT*, the genome sequence showed three tandemly duplicated *Methyltransferase* genes (*Solyc09g091530-50*). The three genes were found in all accessions and the locus did not show extensive SV except for a single 255 bp deletion (SL4.0ch09: 66,899,792–66,900,047) in the promoter of *SlSAMT1* (*Solyc09g091550*) that was found in 27 accessions. The *SAMT1* (*Solyc09g091550*) is known to regulate methyl salicylate levels in tomato fruit [32]. Since we did not identify any other large structural variants, we proceeded with the haplotype analysis using SNPs and small INDELs at the 21.3 kb locus (S5 Fig and S5 Table). A total of 530 variants were identified and were comprised of 399 SNPs and 131 INDELs. Among them, 83% of the

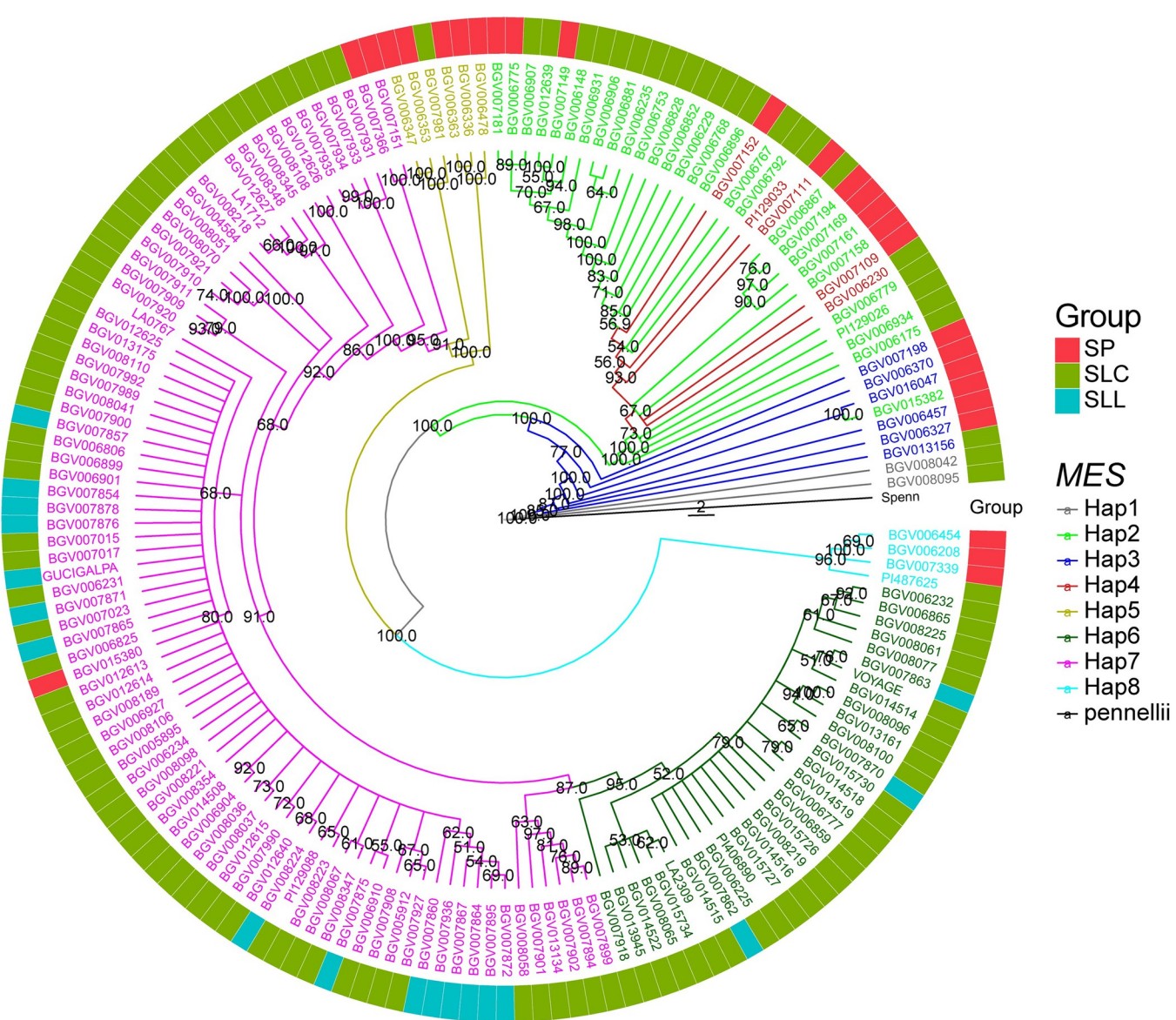

**Fig 6. Phylogeny tree constructed using consensus genomic sequence of *MES* locus and flanking regions from the Varitome Collection along with green fruited *S. pennellii* (Spenn).** Different colors of accessions and branches represent different *MES* haplotypes. The outer concentric circle represents the grouping of accessions (SP, SLC and SLL). The numbers on the branches represent the bootstrap values. *S. pennellii* is used as outgroup.

variants were found in regulatory regions. Of the coding region variants, the known *SAMT1* gene *Solyc09g091550* carried two synonymous and seven missense mutations compared to the Heinz 1706 reference genome for which one or more were found in 34 accessions. This means that the majority of red-fruited accessions in this study encoded the same SAMT1 protein sequence as Heinz1706. For another *SAMT* gene, *Solyc09g091530*, ten synonymous, nine missense, two frameshift, and two nonsense mutations were found compared with the Heinz1706 reference genome. The frameshift and nonsense mutations resulted in null alleles that were found in seven SP, three SLC and one SLL. Lastly, we found six synonymous, seven missense and one frameshift mutation in *Solyc09g091540*. The frameshift mutation was found in only three accessions (2 SLC and 1 SLL) and these also carried the predicted null mutations in

*Solyc09g091530*. The haplotype analysis for *SAMT1* using the 530 variants at the 21.3 kb locus showed four alleles (S5A Fig and S3 and S5 Tables). Haplotype I was found in 19 SP, 13 SLC and three SLL_CUL and had the 255 bp deletion in the upstream regulatory region of *SlSAMT1* as well as six of the 11 accessions with null mutations in *Solyc09g091530*. Haplotype II was found in eight SP and two SLC, including two SP with a premature stop codon in *Solyc09g091530*. Haplotype III appeared in 106 SLC, 18 SLL and 37 SLL_CUL accessions, including three accessions (two SLC and one SLL) with the frameshift mutation in *Solyc09g091530* and *Solyc09g091540*. Haplotype IV was identical to the Heinz1706 reference genome and found in the majority of SLL_CUL. Haplotype I of *SlSAMT* was the most ancestral haplotype based on the phylogenetic tree constructed from the consensus genomic sequence of the locus and included most of the SP accessions (S6 Fig). Other haplotypes, II and III were found to cluster separately under distinct clades in the phylogenetic tree. The phylogenetic analysis did not include haplotype IV since it was found only in SLL_CUL.

At the *UGT5* locus, the only SV was a 403 bp deletion (SL4.0ch01:79,099,508) found in 50 accessions and located 8 kb upstream of the gene. We identified 132 SNPs and 29 small INDELs at the 5.7 kb *UGT5* locus compared to the Heinz1706 reference genome. Most of the variants (150) were found in the regulatory regions and only 11 variants (six missense, four synonymous and one frameshift mutation) were found in the gene. The frameshift mutation was a likely null allele and was found in 41 accessions. Haplotype analysis using the variants in the 5.7 kb region resulted into four distinct haplotypes (S7A Fig and S3 and S6 Tables). Haplotype I was found in nine SP accessions and carried one missense and one synonymous mutation in the *SlUGT5* compared to Heinz reference genome. Haplotype II was found in 18 SP, 23 SLC and two SLL_CUL. All the accessions in this haplotype carried two missense mutations and a frameshift mutation in the gene compared to Heinz1706. Furthermore, the accessions in haplotype I and II also carried the 403 bp deletion. Haplotype III included the Heinz1706 reference genome and was found in 31 SLC, 11 SLL, and 59 SLL_CUL. Haplotype IV and was the most common haplotype and was found in all accessions except SP. Accessions belonging to haplotype IV were similar to Heinz1706 within the gene region whereas Heinz1706 upstream regulatory region was more similar to haplotype III accessions. Phylogenetic analysis showed haplotype II as the most ancestral haplotype (S8 Fig). Haplotype I was clustered separately in the tree followed by a clade with both haplotypes III and IV.

## Association of the haplotypes at *MES*, *NSGT1*, *SAMT1* and *UGT5* loci with methyl salicylate levels

We evaluated the association of the haplotypes at these the loci with methyl salicylate levels, starting with *MES* and *NSGT1*. Among the nine *MES* haplotypes, haplotype 2 produced higher levels of methyl salicylate than the other haplotypes suggesting a loss-of-function allele (Fig 7A). Haplotype 4 and haplotype 9 also produced relatively high levels of methyl salicylate (Haplotype 4: PI129033: 16.61 ng/gfw/hr and haplotype 9: TS-209: 29.36 ng/gfw/hr), however these haplotypes represented only a few accessions (n = 5 in haplotype 4 and n = 13 in haplotype 9). Most accessions with haplotypes 1, 3, 5, 6, 7, or 8 produced low levels of methyl salicylate, with exceptions, suggesting the presence of functional alleles that converted the volatile into salicylic acid. The variation at *NSGT1* is represented by five haplotypes [38]. These haplotypes are classified in two groups based on expected functionality: the functional *NSGT1* haplotypes I, II and III and the non-functional *NSGT1* haplotypes IV and V. The non-functional *NSGT1* haplotypes produced significantly higher levels of methyl salicylate compared to the functional *NSGT1* haplotype (Fig 7B). The combined effect of the *NSGT1* and *MES* haplotypes showed a strong interaction between the loci. *MES* haplotype 2 and a non-functional *NSGT1*

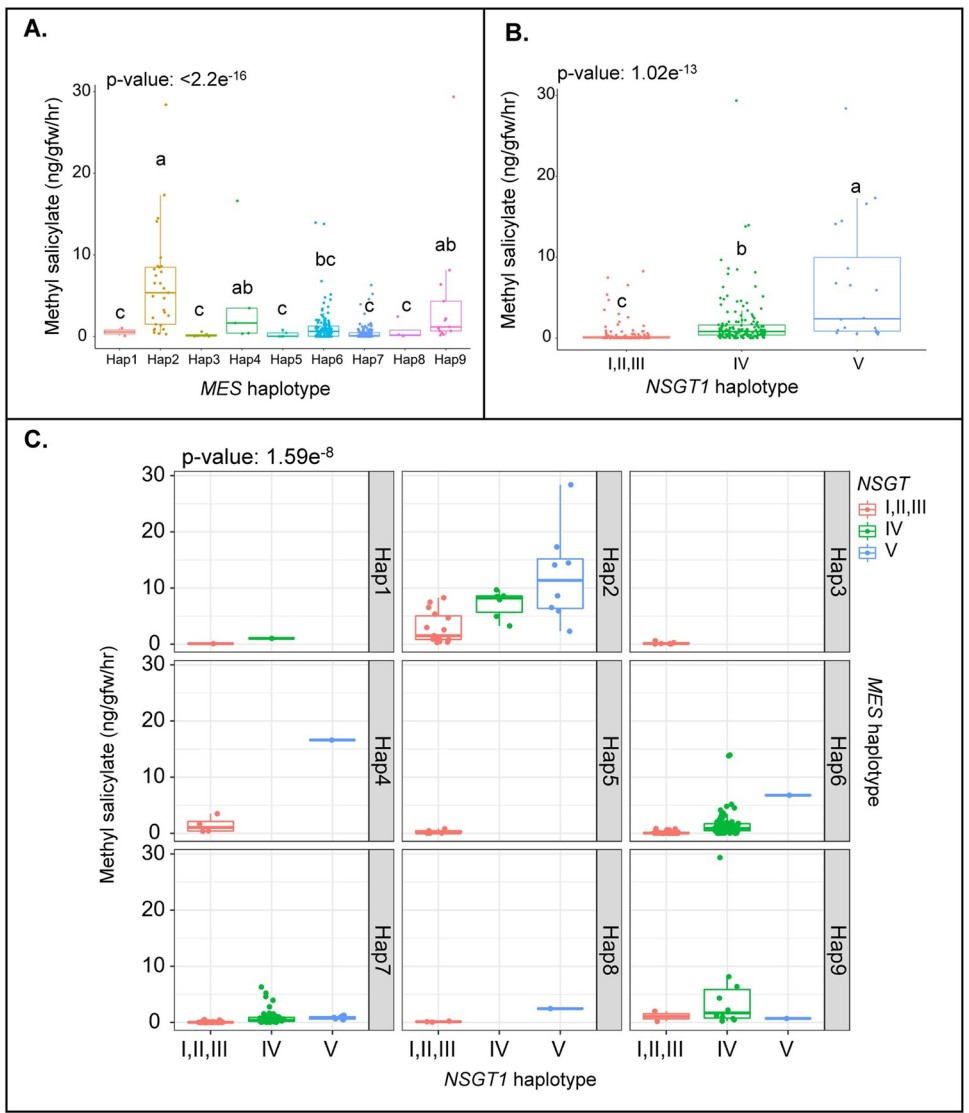

**Fig 7. Distribution of methyl salicylate in the Varitome Collection and cultivated tomato.** (**A**) *MES* haplotypes, (**B**) *NSGT1* haplotypes, and (**C**) across different *MES* and *NSGT1* haplotypes. The letters in the boxplots indicate the significant differences among different haplotypes evaluated by Duncan's test ($\alpha < 0.05$).

haplotype IV or V resulted in the highest methyl salicylate levels (Fig 7C). Previous results from a biparental cross also supported the notion that *MES* and *NSGT1* strongly interact with respect to the methyl salicylate levels [35]. Other haplotype combinations carried too few accessions to reliably ascertain association with volatile levels, although a trend between haplotype 4 and *NSGT1* V suggested that this *MES* haplotype was non-functional.

The four *SAMT1* haplotypes were not statistically different in methyl salicylate accumulation (S5B Fig). Even so, *SAMT1* haplotype II seemed to produce higher levels of methyl salicylate but was represented by too few accessions to significantly associate volatile levels with the haplotype. The 255 bp deletion showed no association with the volatile levels and thus was unlikely causative (S5C Fig). Interestingly, the 11 accessions carrying a null mutation in *Solyc09g091530* were found to produce low levels of methyl salicylate contrary to expectations (S5D Fig). Similar to *SAMT1*, there was no significant difference among the *UGT5* haplotypes

(S7B Fig). *UGT5* haplotype I showed a trend of higher methyl salicylate levels but was found in only nine accessions which is too few for a meaningful analysis. We also did not find a significant association of the 403 bp deletion with the methyl salicylate levels in fruits (S7C Fig). Overall, we could not show association of a causative variant at *SAMT1* and *UGT5* with methyl salicylate levels. This may be due in part to the large effect of *MES* and *NSGT1* on volatile levels, and the relatively low number of accessions that carried the same haplotype at the two major loci to allow a thorough characterization of the two minor loci.

### Expression of *SlMES1-4* in *MES* locus

All haplotypes at the *MES* locus, except haplotype 9, carried putative functional copies of one to several *MES* genes. Especially *SlMES1*, in all accessions except haplotype 9, appeared to be full length and functional (Fig 5). Since the function of *SlMES1* is known [35], we sought to investigate the putative role of other members at the locus in methyl salicylate production in ripe tomato fruit. We investigated their expression by Real-Time Quantitative Reverse Transcription Polymerase Chain Reaction (qRT-PCR) (Fig 8). We selected 37 Varitome accessions that represented eight *MES* haplotypes (Haplotype 1–8). In accessions belonging to haplotype 1 and 3, *SlMES3* was well expressed in ripe tomato fruit, whereas the expression of the other *MES* genes was low or undetectable. Accessions belonging to these haplotypes produced very low amounts of methyl salicylate, ranging from 0.03 to 1.03 ng/gfw/hr (Fig 7A), implying a functional allele. In haplotype 2, 4, and 8, the expression of all four *MES* genes was low to nondetectable. Especially haplotypes 2 and 4 were associated with high accumulation of methyl

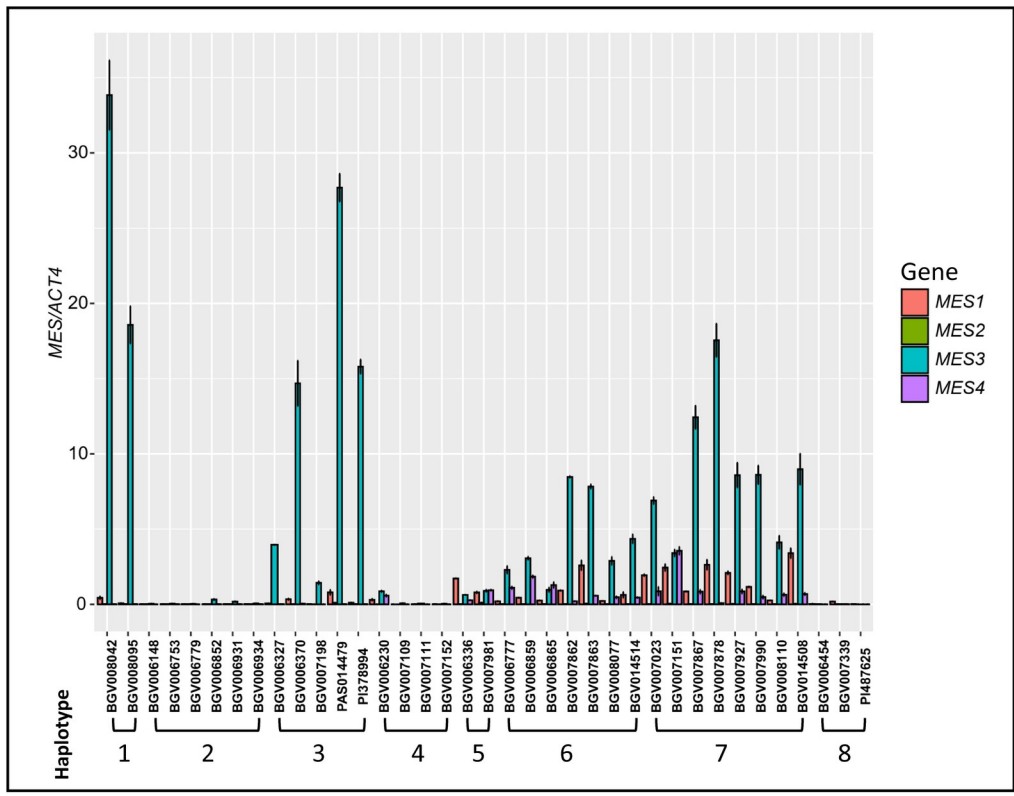

**Fig 8. Relative gene expression of *MES1-4* in different Varitome accessions across the *MES* haplotypes.** The bar diagram represents the mean expression normalized by *ACT4* expression with ± standard deviation. The number below the accessions represents the *MES* haplotype.

**Table 1. Summary of *MES* haplotypes and gene expression.** "+" represents the gene is highly expressed and "-" represents either low or no gene expression. *For haplotype 9, expression levels were predicted based on genome structure of the haplotype.

| Haplotype | | *MES1* | *MES2* | *MES3* | *MES4* | Average MeSA (ng/gmfw/hr) |
|---|---|---|---|---|---|---|
| 1 | Expression | + | - | + | - | 0.562 |
| | mutation | | | | 48 bp del in first exon | |
| 2 | Expression | - | - | - | - | 6.383 |
| | mutation | 363 bp del in promoter | | SNP at first exon | 48 bp del in first exon | |
| 3 | Expression | + | - | + | - | 0.191 |
| | mutation | | | | 48 bp del in first exon | |
| 4 | Expression | - | - | - | - | 4.506 |
| | mutation | 363 bp del in promoter | | SNP at first exon | 48 bp del in first exon | |
| 5 | Expression | + | - | + | + | 0.273 |
| | mutation | | | | | |
| 6 | Expression | + | - | + | + | 1.135 |
| | mutation | | | | | |
| 7 | Expression | + | - | + | + | 0.479 |
| | mutation | | | | | |
| 8 | Expression | - | - | - | - | 0.720 |
| | mutation | | | deletion | 48 bp del in first exon | |
| 9* | Expression | - | - | - | - | 4.378 |
| | mutation | deletion | deletion | deletion | deletion | |

salicylate, likely due to the lack of expression of any of the *MES* genes. The two accessions representing haplotype 5 showed slightly higher expression of *SlMES1*, *SlMES3* and *SlMES4*, compared to haplotype 2, 4, and 8. In general for haplotype 6 and 7, *SlMES3* was highly expressed followed by *SlMES1* and last *SlMES4*. The expression of *SlMES2* was undetectable in nearly all accessions suggesting that this gene was not important for methyl salicylate accumulation in the fruit. As expected, the expression patterns of *MES* were negatively correlated with the fruit methyl salicylate levels. Haplotypes producing high level of methyl salicylate: haplotype 2 and 4, showed low to non-detectable expression of *MES* genes. On the other hand, haplotypes that produced low levels of methyl salicylate, haplotypes 1, 3, 6 and 7, *SlMES1* and/or *SlMES3* were highly expressed in ripe fruits. A summary of *MES* haplotypes, gene structure, and expression is shown in Table 1.

## Evolution of the *MES* and *NSGT1* loci during tomato domestication and selection

SP and SLC accessions that were collected in Ecuador showed higher levels of methyl salicylate whereas SLL and most of SLL_CUL featured low levels of the volatile (Fig 2B). This suggested selection by the terrestrial ecosystem and/or domestication on volatile production, presumably through the *MES* and *NSGT1* loci. To investigate how these two loci were selected during the evolution of tomato, we evaluated the haplotype distribution in the Varitome Collection and cultivated accessions (Fig 9). Most of the SLL_CUL, the majority of SLL, and many SLC carried the *MES* haplotype 6 and 7, suggesting strong selection for these alleles during domestication (Fig 9A) as these haplotypes were found to produce low levels of methyl salicylate (Fig 7A). The distribution map also showed that haplotypes 6 and 7 spread across several regions from Ecuador, Peru, to Colombia and Mexico. Not surprisingly, the most common allele in Ecuador was *MES* haplotype 2, associated with high levels of methyl salicylate. For *NSGT1*, we found the nonfunctional haplotypes IV and V to be the most common haplotypes in

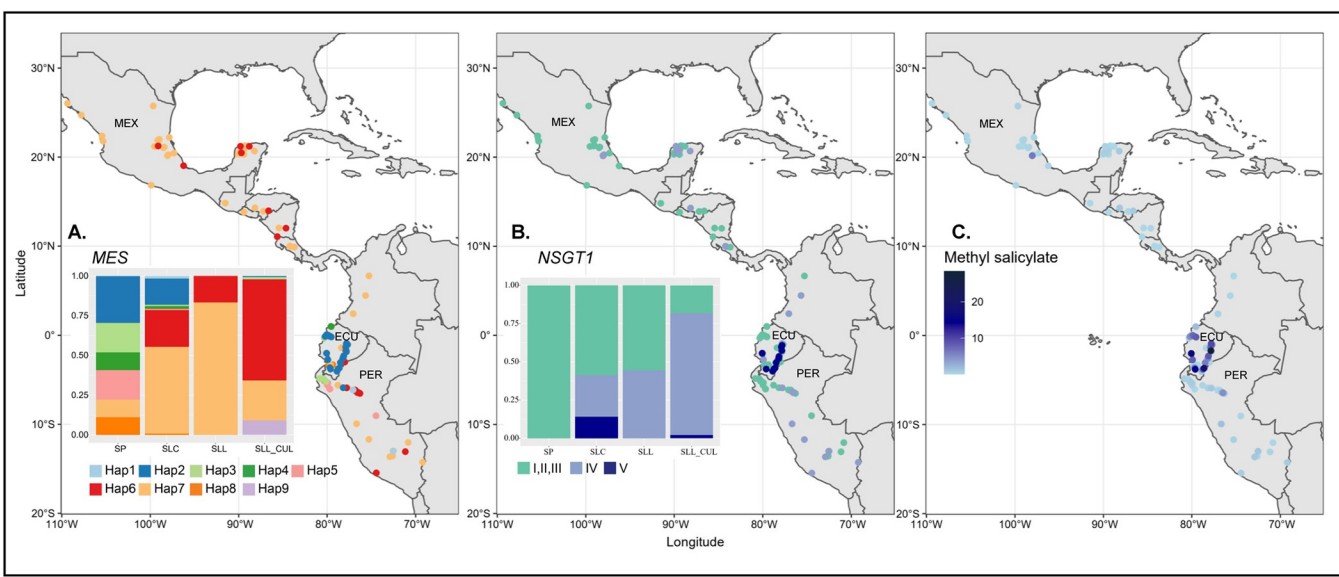

**Fig 9.** Proportion bar stack plot and distribution map of accessions in Varitome Collection (A) *MES* haplotypes (B) *NSGT1* haplotypes. (**C**) Distribution map of methyl salicylate levels in the accessions. The base layer of the map was generated using R package "rnaturalearth", which uses the public domain map dataset "Natural Earth" (https://www.naturalearthdata.com/).

SLL_CUL. The proportion of accessions belonging to *NSGT1* haplotype IV and V was found to increase from SP (0%) to SLC (41.32%), SLL (44.44%) and SLL_CUL (81.81%) (Fig 9B). Based on the distribution map, haplotype I, II, III had spread across several regions in South and Central America whereas haplotype V was found only in Ecuador and northern Peru. Together, these data suggested that functional *NSGT1* (Haplotype I, II, III) were under selection in SP but not in cultivated tomato, whereas the opposite was true for *MES*. For the two loci combined, it appeared that a strong selection for higher levels of methyl salicylate were particularly important to certain SP and SLC accessions from Ecuador and Peru (Fig 9C), with a non-functional haplotype for both *MES* and *NSGT1* (Fig 9A and 9B). The high levels of methyl salicylate might have contributed to the survival in the specific ecosystem that is common to that region.

## Discussion

Methyl salicylate is an important plant volatile that functions as a signaling molecule within and between plants. Its role in defense responses in leaves is relatively well understood. It serves as an herbivore and pathogen-induced volatile that attracts parasitoids and predators of the herbivores. Plants emit methyl salicylate when attacked by herbivores, sending cues to parasitoids and other predators [11,39–41]. Furthermore, methyl salicylate acts as a signaling molecule to activate the genes that are important in regulating the plants' response to biotic and abiotic stress [35,42–44]. Methyl salicylate also functions as an inter-plant signal that activates disease resistance and the expression of defense-related genes in surrounding plants [39,43]. Even though its role in defense responses is well established, the function of methyl salicylate in ripe fruit is not well understood. We show that the levels of this volatile vary dramatically in red-fruited tomato fruits and are particularly high in accessions originating from Ecuador. This suggests that methyl salicylate in ripe fruits may be beneficial to plants and that certain levels of this volatile in tomato are acceptable to consumers. It is likely that high levels of methyl salicylate in ripe fruits could serve as either an attractant to frugivores or deterrent to

insect pests that often function as vectors of pathogenic microbes and viruses. No studies have been conducted on the effect of methyl salicylate levels on fruit health, but external applications and pretreatment of this volatile have been reported to reduce post-harvest losses in tomatoes. Methyl salicylate application reduces chilling injury [45] alleviates post-harvest fungal decay [46,47], and suppresses loss of several desirable aromatic compounds [48]. Hence, even though the volatile is associated with low consumer liking, methyl salicylate could be important in reducing the impact of biotic and abiotic stressors on the plant and plant products such as the fruit.

*MES*, along with *NSGT1* is a major contributor to fruit methyl salicylate levels in the tomato germplasm. Moreover, the interaction between *MES* and *NSGT1* explained most of the variation in fruit volatile levels in a biparental mapping population [35], supporting the findings from this study. The combination of *MES* haplotype 2 and a non-functional *NSGT1* haplotype IV or V resulted in the highest methyl salicylate levels indicating that the interaction of these two loci leads to higher levels of this volatile. Other genes in this pathway, *SlSAMT1* and *SlUGT5*, appeared as minor players in this germplasm. The high or low methyl salicylate level haplogroups of *SAMT1* and *UGT5* could be explained by their genotype at *MES* and *NSGT1* loci. *SAMT1* haplotype II seemed to produce high levels of methyl salicylate. All accessions with *SAMT1* haplotype II carried a non-functional *MES* haplotype (haplotype 2 or 4) which might explain the high volatile levels even though the production of methyl salicylate is dependent on a functional *SAMT1*. In addition, the 11 accessions carrying a null mutation in *Solyc09g091530*, whose function in methyl salicylate production is unknown, carried the functional *NSGT1* haplotype which could explain the low volatile levels in the fruit. It is likely that there are other fruit-specific glycosyltransferases and methyltransferases with activity on methyl salicylate and salicylic acid respectively, downplaying individual effect of the cloned *SlSAMT1* and *SlUGT5* [32,33,49]. Additionally, too few accessions remained after considering the effect of the major two loci (*MES* and *NSGT1*) to conclude which was the most likely active or inactive allele at the minor loci.

The *MES* and *NSGT1* loci have been shaped predominantly by SVs. It is known that SVs greatly affect gene expression and protein functions resulting in variations in phenotype [38,50]. *MES* carries several deletions and insertions, spanning the four *Methyltransferase* genes either partially or completely deleting genes from the locus. These SVs were likely responsible in part for the difference in *MES* gene expression observed among the haplotypes (Table 1). *SlMES1* appeared to be the only functional gene in all haplotypes, except for haplotype 9. However, the expression of this gene was very low or undetected in haplotypes carrying the 363 bp deletion in the promoter of *SlMES1* but was higher for haplotype 8 which did not have the deletion. *SlMES3* was the highest expressed gene at the *MES* locus especially in haplotypes 1, 3, 5, 6 and 7. We did not find a correlation between gene expression and transposon insertion or truncation of *SlMES3*. Instead, we identified a SNP (SL4.0ch02: 34,463,357; C to T) in the first exon of *SlMES3* found only in accessions in haplotypes 2 and 4. This SNP is predicted to cause an amino acid change from leucine to phenylalanine at position 83, which is one of the salicylic acid binding sites of the protein [51] and possibly consequential. This finding suggests that *SlMES3* is likely functional in addition to *SlMES1*. For *SlMES4*, expression levels were associated with the truncation of the protein. Haplotypes 1, 2, 3, 4, and 8 carrying a truncated *SlMES4* had low or undetectable expression, whereas the haplotypes 5, 6, and 7 carried full length *SlMES4* that showed high expression.

Several events in selection and domestication of cultivated tomatoes have affected the fruit methyl salicylate levels and *MES* controlling the production of this volatile. Recent evolutionary models proposed by Razifard et al. [28] and Blanca et al. [29] can be compared to the *MES* haplotype distribution. SP from Peru, considered as the most diverse and ancestral, includes

**Fig 10.** Linear illustration of evolutionary model proposed by **A**. Blanca et al. [29] and **B**. Razifard et al. [28]. The numbers in blue represent the *MES* haplotypes. The gray solid arrow represents the direction of evolution. The abbreviations are PER: Peru, ECU: Ecuador, and MA: Mesoamerica. SLC from PER and ECU includes some accessions which are admixture of SP and SLC.

accessions with several *MES* haplotypes (Fig 10). During its spread north, the tomato plants would have encountered biotic and abiotic conditions associated with higher latitudes, such as changes in temperature, water availability, or disease, which likely shaped the natural selection for *MES* haplotypes. SP from southern Ecuador carried the functional *MES* haplotype (7), while SP from northern Ecuador carried the non-functional haplotypes (2 and 4). Non-functional *MES* haplotype and high levels of methyl salicylate may have contributed to their survival in adverse environments. It is also possible that birds or animal dispersers, having preference for minty flavor, contributed to selection and migration, but little is known about this. However, selections imposed by the new environment necessitates adaptation that might have favored the selection of a specific *MES* haplotype over the other. According to Blanca et al. [29], SP spread into Mesoamerica where it evolved into SLC and the ancestral SLC migrated back to South America where it hybridized with native SP populations. Following this model, the most functional *MES* haplotype 7, with potentially four functional genes, was selected as Peruvian and Ecuadorian SP evolved into SLC in Mesoamerica. *MES* haplotype 6 evolved from haplotype 7, after a premature stop-codon mutation in *MES2*. Upon migration back to South America, admixture with Ecuadorian SP resulted in SLC from Ecuador carrying the most diverse *MES* haplotypes. SLC Peru evolved from admixture with Peruvian SP resulting in at least four *MES* haplotypes, and together with Ecuadorian SLC spread northwards to evolve into ancestral SLL carrying only haplotype 6 and 7. On the other hand, Razifard et al. [28] proposed that SP from northern Ecuador evolved into SLC in South America carrying several domestication alleles which then migrated to Mesoamerica where it evolved into SLL. Based on this model, the functional *MES* haplotype 7 and its derivative haplotype 6 are found in Peruvian and Ecuadorian SP and were selected early as tomato migrated northwards to evolve into SLL. The evolution in *MES* locus seems to agree more with the model proposed by Razifard et al. [28] as the linear gene flow model from Peruvian and Ecuadorian SP to Mexican SLL explained the decrease in diversity at the locus. Regardless of the model, large genetic diversity at the *MES* locus was maintained in South America whereas migration northwards into Mesoamerican SLC and eventually SLL resulted in the selection of only two *MES* haplotypes 6 and 7. Thus, the evolution at the *MES* locus suggests a complex migratory and evolutionary path of tomato.

The genetic variation for the known genes associated with methyl salicylate biosynthesis and conjugation in the Varitome Collection was large, especially at the *MES* locus. Moreover,

even within each distinct haplotype, we observed wide phenotypic variation (Fig 7), suggesting that additional genetic factors control the biosynthesis process. GWAS using the Varitome Collection identified several additional QTL associated with fruit methyl salicylate levels in addition to known genes (Fig 3 and S2 Table). These QTL could harbor novel genes, either affecting the homeostasis of methyl salicylate including additional *Glycosyltransferases* and *Glucosidases*. The Varitome Collection thus, is a valuable tool to discover additional genes associated with volatile production through a genetic mapping approach. Identification of the novel genes would allow us to gain a deeper understanding of the regulation of methyl salicylate in the fruits.

The results presented herein contribute to understanding the genetics of the biosynthesis and conjugation of methyl salicylate in ripe tomato fruits. The beneficial haplotypes at the *MES* and *NSGT1* loci could be introgressed into cultivated germplasm to obtain optimal flavor depending on the agroecosystem. Introgression of beneficial haplotypes of *NSGT1* has the added advantage of decreasing levels of another undesirable volatile, guaiacol, thereby improving flavor. Thus, the genomic information from this study can be used to develop molecular tools to perform marker-assisted selection for the desirable haplotype at each locus. Intermediate levels of methyl salicylate in modern and heirloom cultivars suggest that certain amounts of this volatile are acceptable or were inadvertently selected because of a strong breeding focus on yield and disease resistance. If the latter, breeding that is focused on the optimal allele selection that favors flavor is now an attainable goal. Thus, the genetic diversity knowledge can be used to fine-tune the methyl salicylate levels to arrive at an acceptable amount of methyl salicylate for overall plant health and resistance as well as consumer satisfaction to balance the desired taste with other post-harvest traits such as shelf life.

## Methods

### Plant materials and phenotyping

A panel of 166 tomato accessions, the Varitome Collection, consisting of 27 wild SP, 121 semi domesticated SLC and 18 ancestral landrace SLL, as well as 143 heirloom and modern varieties (SLL_CUL) were grown and phenotyped for volatile accumulation as previously described [25,28]. Based on collection site and whole genome information, the species that comprise these accessions were further divided into three SP subpopulations (SP_SECU from Southern Ecuador, SP_NECU from Norther Ecuador and SP_PER from Peru); six SLC subpopulations (SLC_ECU from Ecuador, SLC_PER from Peru, SLC_SM from San Martin region of Peru, SLC_CA from Central, Northern South and Southern North America, SLC_MEX from Mexico, and SLC_ADM which are admixture SLC accessions); and two SLL subpopulations namely SLL comprised of ancestral landraces and SLL_CUL comprised of heirloom and modern accessions. The red ripe fruits were harvested, cut, and loaded in glass tubes. Air was passed over samples for an hour and the volatiles were collected on a SuperQ Resin column. Volatiles were then eluted off the column with methylene chloride and ran on GC-MS for analysis as previously described [25]. The volatile measurements were average of volatile levels from at least two and up to five harvests.

### Statistical analysis

All statistical analysis were conducted using R [52] unless mentioned otherwise. The figures were generated using "ggplot2" package [53] in R. The multiple mean comparison was performed using the "duncan.test" function from "agricolae" package in R [54].

## Variant calling and genome-wide association study

Raw ILLUMINA read files were downloaded from NCBI (https://www.ncbi.nlm.nih.gov/; SRA: SRP150040, SRP045767, SRP094624, and PRJNA353161). We augmented the original GWAS by Razifard et al. [28] by aligning the reads to the most recent tomato genome SL4.0 build as opposed to SL2.50. We also interrogated the three main variant types, namely SNPs, INDELs and SVs for the GWAS compared to SNPs only [35]. Raw read evaluation, filtering, alignment, and variant calling were performed as previously described [55]. The phenotype deviating from normality (p value from Shapiro test <0.01) was normalized using quantile normalization in R (52). Associations between genotype and phenotype were calculated using the FarmCPU model in GAPIT (version 3) [56] following the established protocols [35]. Association analysis was conducted using 21,893,681 SNPs, 2,735,297 INDELs and 27,477 SVs. The association results were plotted using the "ggplot2" package in R [53].

## Haplotype analyses

To identify the haplotypes at the known genes regulating methyl salicylate levels, we first determined the presence of SVs at or near each by comparing the high-quality genome sequences of 28 Varitome accessions (ftp://ftp.solgenomics.net/genomes/tomato100/March_02_2020_sv_landscape/variants/). The genomes of these accessions were sequenced using Oxford Nanopore Technology and Illumina short read sequencing and, therefore, were well suited for the identification of SV at each of the loci. For the loci with evidence of SV, *MES* and *NSGT1*, SVs were identified in Integrative Genome Viewer (IGV) [57] on the pattern of short reads in the tomato accessions, and haplotypes were defined largely based on SVs. The size of the *MES* locus ranged from 15 kb to 35 kb, whereas the size of the *NSGT1* locus was as defined previously [38]. For *SlSAMT1*, the locus comprised three tandem duplicated genes without large SVs. To define the haplotypes at the *SAMT1* and *UGT5* loci, SNPs and small INDELs within the coding region, 3 kb upstream and 1 kb downstream of the termination of transcription were extracted using VCFtools [58]. The method to generate the haplotypes based on SNPs and INDELs was described previously [55]. Variants (SNPs and INDELs) were annotated using SnpEff [59] using a locally built database of SL4.0 tomato reference genome. The FGE-NESH [60] tool was used to verify and predict gene models. The haplotype distribution map was created using R package "rnaturalearth" (https://CRAN.R-project.org/package=rnaturalearth) and "ggspatial" (https://CRAN.R-project.org/package=ggspatial). The package "rnaturalearth" uses the public domain map dataset "Natural Earth" (https://www.naturalearthdata.com/) to generate the base layer of the map.

## Phylogenetic tree construction of the *Methylesterases* at *MES* locus

A phylogenetic tree was built using Neighbor-Joining method with the genomic sequences of genes at the *MES* locus and 10 kb flanking the locus. The consensus nucleotide sequences were generated using IGV and, exported in Geneious Prime v2021.2.2 (https://www.geneious.com/). The sequences were then aligned using the Clustal alignment option. The tree was constructed using Neighbor-Joining method and Tamura-Nei genetic distance model using *S. pennellii* (sequence obtained from SGN; https://solgenomics.net/) as an outgroup. Bootstrap support for the tree was obtained using 100 bootstrap replicates. The tree was then plotted using "ggtree" package in R [61].

## qRT-PCR

qRT-PCR was performed following a protocol that was used previously [35]. Total RNA was extracted from red ripe fruits of 37 accessions and used for the qRT-PCR, including some that were collected before [35]. *ACTIN4* (*Solyc04g011500*) was used as the control to standardize relative expression levels. The relative expression was calculated using the formula of ΔCq method using a reference gene: R = 2Cq(reference)-Cq(target), where Cq is quantification cycle, reference is *ACT4*, and targets are *MES1*, *MES2*, *MES3*, or *MES4*. The expression analysis figure was generated using "ggplot2" package in R [53]. The sequences of primers used in the study are provided in S7 Table.

## Data access

All the raw sequencing data used in this study are publicly available in NCBI (https://www.ncbi.nlm.nih.gov/; SRA: SRP150040, SRP045767, SRP094624, and PRJNA353161).

## Supporting information

**S1 Fig. Genomic alignment of Varitome accessions to SL4.0 genome version at *MES* locus.** (A) Alignment of three highest methyl salicylate producing accessions against the SL4.0 genome build at the *MES* locus. (B) Alignment of three least methyl salicylate producing accessions against the SL4.0 genome build at the *MES* locus. Yellow boxes represent gene models. (PDF)

**S2 Fig. ClustwalW alignment of predicted protein sequence of *SlMES1-4* of all *MES* haplotypes.** Alignments of (A) *SlMES1* (B) *SlMES2* (C) *SlMES3* and (D) *SlMES4* (PDF)

**S3 Fig. Comparison of *MES* haplotype 2 and 5 and representation of a transposon insertions between *SlMES2* and *SlMES3; and within *SlMES3*.** The gene model below is a screenshot from SL4.0 genome version of the exact region which is the insertion showing a transposon insertion between *SlMES2* and *SlMES3*. (PDF)

**S4 Fig. Alignment of Rio Grande against the SL4.0 genome build at the *MES* locus.** Alignment for *SlMES1*, *SlMES2*, *SlMES3 and SlMES4*. Yellow boxes represent gene models. (PDF)

**S5 Fig. Haplotype analysis of *SAMT1* locus.** (A) Heatmap representing the genotypes of accessions (rows) for the polymorphisms identified (columns). Reference genotypes are represented in blue, alternate in red, heterozygous in yellow and missing data in white. The black rectangular box represents the position of the genes in the locus. (B) Distribution of methyl salicylate in red fruits in different accessions among different *SAMT1* haplotypes. (C) Distribution of methyl salicylate in red fruits in different accessions with and without deletion in the *SAMT1* locus. (D) Distribution of methyl salicylate between accessions with and without null mutations of *Solyc09g091530*. (PDF)

**S6 Fig. Phylogeny tree constructed using consensus DNA sequence of *SAMT1* locus and 10 kb flanking region of *SAMT1* locus from Varitome collection.** Different colors of accessions and branches represent different *SAMT1* haplotypes. The outer two concentric circles represent the grouping of accessions (SP, SLC and SLL) and corresponding methyl salicylate levels (MeSA) in the accessions respectively. The numbers on the branches represent the bootstrap

values.
(PDF)

**S7 Fig. Haplotype analysis of *UGT5* locus.** (A) Heatmap representing the genotypes of accessions (rows) for the polymorphisms identified (columns). Reference genotypes are represented in blue, alternate in red, heterozygous in yellow and missing data in white. The black rectangular box represents the position of the gene in the locus. (B) Distribution of methyl salicylate in red fruits in different accessions among different *UGT5* haplotypes. (C) Distribution of methyl salicylate in red fruits in different accessions with and without deletion in the promoter of *SlUGT5*.
(PDF)

**S8 Fig. Phylogeny tree constructed using consensus DNA sequence of *UGT5* locus and 10 kb flanking region of *UGT5* locus from Varitome collection.** Different colors of accessions and branches represent different *UGT5* haplotypes. The outer two concentric circles represent the grouping of accessions (SP, SLC and SLL) and corresponding methyl salicylate levels (MeSA) in the accessions respectively. The numbers on the branches represent the bootstrap values.
(PDF)

**S1 Table. List of significant loci associated with methyl salicylate levels and their respective variants from SNP, INDEL and SV GWAS, their position, p-value, LOD values and cloned genes.**
(XLSX)

**S2 Table. List of SVs identified in *MES* locus and used for defining *MES* haplotypes.**
(XLSX)

**S3 Table. Methyl salicylate levels (ng/gfw/hr) in Varitome collection and heirlooms and their respective *MES* and *NSGT1* haplotype.**
(XLSX)

**S4 Table. Annotation of all the variants (SNPs and INDELs) in the *MES* locus in Varitome collection.** The *MES* haplotype information is provided in the column besides the accession names. The annotation information is provided along each variant. 0|0 represents reference allele, 1|1 represents alternate allele, and 1|0 represents heterozygous allele. The yellow highlighted variants are the variants discussed in the text.
(XLSX)

**S5 Table. Annotation of all the variants (SNPs and INDELs) in the *SAMT1* locus in Varitome collection.** The *SAMT1* haplotype information is provided in the column besides the accession names. The annotation information is provided along each variant. 0|0 represents reference allele, 1|1 represents alternate allele, and 1|0 represents heterozygous allele. The yellow highlighted variants are the variants discussed in the text.
(XLSX)

**S6 Table. Annotation of all the variants (SNPs and INDELs) in the *UGT5* locus in Varitome collection.** The *UGT5* haplotype information is provided in the column besides the accession names. The annotation information is provided along each variant. 0|0 represents reference allele, 1|1 represents alternate allele, and 1|0 represents heterozygous allele. The yellow highlighted variants are the variants discussed in the text.
(XLSX)

**S7 Table. qRT PCR primers for the expression analysis of *SlMES1-4* and *ACT4*.**
(XLSX)

**S8 Table. Raw data to generate Figures 4, 6, 8, and 9.** The data are presented in different worksheets.
(XLSX)

## Acknowledgments

We thank the Georgia Advanced Computing Resource Center, UGA, for providing the computation resources.

## Author Contributions

**Conceptualization:** Manoj Sapkota, Lara Pereira, Esther van der Knaap.

**Data curation:** Manoj Sapkota.

**Formal analysis:** Manoj Sapkota.

**Funding acquisition:** Esther van der Knaap.

**Investigation:** Manoj Sapkota, Lara Pereira, Yanbing Wang, Yasin Topcu, Esther van der Knaap.

**Methodology:** Manoj Sapkota, Lara Pereira, Yanbing Wang, Yasin Topcu, Denise Tieman, Esther van der Knaap.

**Project administration:** Esther van der Knaap.

**Resources:** Denise Tieman, Esther van der Knaap.

**Software:** Manoj Sapkota.

**Supervision:** Lara Pereira, Lei Zhang, Denise Tieman, Esther van der Knaap.

**Validation:** Manoj Sapkota.

**Visualization:** Manoj Sapkota, Yanbing Wang, Yasin Topcu.

**Writing – original draft:** Manoj Sapkota, Esther van der Knaap.

**Writing – review & editing:** Manoj Sapkota, Lara Pereira, Yanbing Wang, Lei Zhang, Yasin Topcu, Denise Tieman, Esther van der Knaap.

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
