## [Decision Letter · Decision Letter 0]

23 Mar 2023

Dear Dr van der Knaap,

Thank you very much for submitting your Research Article entitled 'Structural variation underlies functional diversity at methyl salicylate loci in tomato' to PLOS Genetics.

The manuscript was fully evaluated at the editorial level and by two independent peer reviewers. We apologize for the lengthy amount of time that this paper was in the review process - we had a difficult time securing the necessary reviews.  The reviewers appreciated the attention to an important topic but identified some concerns that we ask you address in a revised manuscript. Neither of the reviewers had major concerns or large suggestions for additional work that are necessary.  However, both of the reviewers make a number of suggestions regarding the presentation or interpretation of the results that could be quite helpful towards improving the manuscript. 

We therefore ask you to modify the manuscript according to the review recommendations. Your revisions should address the specific points made by each reviewer.

Yours sincerely,

Nathan M. Springer

Academic Editor

PLOS Genetics

Li-Jia Qu

Section Editor

PLOS Genetics

Reviewer's Responses to Questions

**Comments to the Authors:**

Reviewer #1: A manuscript entitled “Structural variation underlies functional diversity at methyl salicylate loci in tomato” investigates how the allelic variation of four genetic loci shape the quantitative variation of methyl salicylate in tomato fruit and how this variation might have evolved during evolution and domestication. The manuscript contains a large amount of novel interesting information, that can prove useful for both fundamental understanding of metabolic variation in tomato fruit as well as for applied proposes, such as breeding for specific chemical composition of tomato fruit. The quality of the study and the presentation of the results is high, but there are a few points that I’d like to see addressed.

Line 80 “While the levels of methyl salicylate are generally low in red-fruited tomato including heirlooms, some produce intermediate levels of the volatile in ripe fruit, suggesting that selections for yield and disease resistance were at the expense of better flavor, a common concern among consumers” – selection for yield and resistance is indeed often blamed for the poor flavor of many modern tomato cultivars. However, I don’t immediately see how varying levels of MeSa suggest that “selections for yield and disease resistance were at the expense of better flavor”. The data present in literature suggests that the ratio between relatively high and low MeSA tomatoes is roughly 50/50 and fruit type independent. Wouldn’t this suggest that the MeSA fruit phenotype is rather neutral for both flavor and tomato plant well-being? Besides, a lower MeSA content would be rather beneficial since a higher content of this volatile often correlates with off-flavor.

Besides that, the importance of MeSA as a selection factor for flavour is questionable. The maximum concentration in the accession collection reported here, which represent a wide genetic diversity, is about 30 ppb whereas the odor threshold reported for MeSA is about 40 ppb (Baldwin et al., 2000, HORTSCIENCE) and this is water meaning that in a fruit matrix it would be even higher. The observation that MeSA has often been associated with off-flavor comes from the correlation and regression analyses, which do not necessarily prove the causal effect. In this particular case the off-flavor could be more likely caused by e.g. guaiacol, which often highly correlates to MeSA (mostly due to NSGT), has a comparable concentration range, but a much lower odor threshold.

Of the four loci studied only NSGT and MES seem to explain certain amount of variation in MeSA. Line 332 states that “Overall, we could not show functional association of SAMT1 and UGT5” - Nevertheless there's a significant association of MeSA and these two loci in the GWAS. Does this mean SAMT1 and UGT5 are not the genes underlying the variation in these loci?

Besides that, accessions with identical MES/NSGT haplotypes show quite different MeSA contents. For instance, in table S3 accession TS-275 - low MeSA - has exactly the same HPT combination as the top producer TS209. Thus, what % variation these do these two loci actually explain? Does this mean there's something else? And why it doesn’t come out of the GWAS?

The discussion on the evolutionary development of the loci is very interesting, but, in my opinion, the authors have put too much weight on MeSA as a driving force of selection for these loci. The effect of the NSGT locus is certainly not restricted to MeSA only, but it also glycosylates other compounds. The same might hold for MES that might act on other ester substrates, which, as recent studies showed, are present in fruits of wild tomato species. They can also demethylate some non-volatile secondary metabolites that have not been analysed in the tomato material studied here.

The picture of both structural alterations and expression levels is kind of scattered throughout the manuscript which makes it difficult to grasp. Can these somehow be summarized (at least for MES and NSGT loci, since the other two don't do much) in e.g. a table or a figure so that a reader could in one glance see what likely to cause a certain MeSA chemotype in different haplotypes, e.g. is that a deleterious mutation in a certain gene(s) or expression or both?

Reviewer #2: Structural variation underlies functional diversity at methyl salicylate loci in tomato.

This is a nice manuscript that takes advantage of the VARITOME collection that has been genotyped and phenotyped by this consortium of partners with the aim to define the origins and evolution of tomato and eventually use of this knowledge in breeding.

In this MS authors focus their attention in the different loci contributing to one of the VOCs that are produced by tomato fruits and generally associated negatively with good tomato flavor MeSA and the authors speculate about a hypothetical agroecological role it could have in the tomato evolution from the center of origin in South and Meso America

The key genes regulating MeSA and their diversity in the tomato clade has been described in a few seminal papers that counted in some with the participation of the authors of this paper. Namely, the key role of the MES cluster and the NSGT have been described in detail from evolution to biochemistry . (MES in Plant Physiology, Volume 191, 110–124; https://doi.org/10.1093/plphys/kiac509 ). NSGT molecular genetics also in Alonge et al (2020) Cell. 2020 Jul 9; 182(1): 145–161.e23. doi: 10.1016/j.cell.2020.05.021

Thus some results seem to me they lack novelty (ie GWAS figures in the PP paper and in the current MS are basi1cally the same) and some of the RT PCR results don’t reveal anything substantially new that was not revealed in the PP paper.

Also de authors describe in the text many details that can be directly obtained from figures or unnecessary.

Authors should cleary indicate what data were generated and original for this ms ad which already reported and indicate were to find the original data as required by PlosGenetics.

To discuss about the “functional” diversity in relation to a putative eco/evolutionary advantage is highly speculative as it is based on single measurements at the ripe stage of fruit (from probably/growing under standard in greenhouse conditions) and the role of MeSA in ripe fuir in relation to these aspects has still to be substantiated with more results. MeSA is known to respond to evironmental cues and the authors know and reported in the PP. This is MeSA-centric ripe fruit-centric paper and when talking about interaction with biotic and abiotic conditions, it misses to include in the analysis SA (the well non mediator if these type of interactions, and the genes involved in the Biosynthesis and metabolism (including SA glycosylation and also misses what happens at the vegetative/plant level to address the question of any evolutionary/ecological role.

Highly speculative: all in this paper is derived from (single) measurements of MeSA in ripe fruit samples so it should not sound like these fruit MeSA related loci can explain ecological adaptations that may depend on plant SA levels for instance more than in red fruit MeSA.

Authors take for granted that the triglycosilated form of MeJA produced by NSGT is an end irreversible product and this appears to be the case in tomato but is it possible that some accessions of SP or SLC may have a specific glycosidases that could make this last step reversible by cleaving the triglyco20sid form of MeSA?

Please give details on how VOCs were determined. If is by cutting the fruits and flow and trap from tomato wedges as in the PP papers is it possible that MeSA production changes during wounding, release and capture and theexpression of corresponding biosynthetic and metabolism genes. If so then could it be that the MES steady state levels obtained by RNA extraction of ripe fruit are missing a possible induction produced in the wounded tomato wedges during MeSA capture and also does not capture the eventual activataion of the MES, NSPGT etc in response to the different environmental conditions that the accessions were experiencing during their evolution.

In summary the story build on the evolution of the traits and the underlying haplotypes is interesting and well constructed ad of interest. Not sure if adds much in comparison to the PP paper about clues for flavou breeding. Technically is ok but I found it highly speculative for the ecological story build around it.

**Have all data underlying the figures and results presented in the manuscript been provided?**

Reviewer #1: Yes

Reviewer #2: **No: **See comment above.

PLOS authors have the option to publish the peer review history of their article (what does this mean?). If published, this will include your full peer review and any attached files.

Reviewer #1: No

Reviewer #2: No

---

## [Editor Report · Decision Letter 1]

19 Apr 2023

Dear Dr van der Knaap,

We are pleased to inform you that your manuscript entitled "Structural variation underlies functional diversity at methyl salicylate loci in tomato" has been editorially accepted for publication in PLOS Genetics. Congratulations!  Thank you for the careful responses to the reviewer comments and suggestions.

Yours sincerely,

Nathan M. Springer

Academic Editor

PLOS Genetics

Li-Jia Qu

Section Editor

PLOS Genetics

Comments from the reviewers (if applicable):

**Data Deposition**

http://datadryad.org/submit?journalID=pgenetics&manu=PGENETICS-D-22-01442R1

**Press Queries**

---

## [Editor Report · Acceptance letter]

29 Apr 2023

PGENETICS-D-22-01442R1 

Structural variation underlies functional diversity at methyl salicylate loci in tomato 

Dear Dr van der Knaap, 

We are pleased to inform you that your manuscript entitled "Structural variation underlies functional diversity at methyl salicylate loci in tomato" has been formally accepted for publication in PLOS Genetics! Your manuscript is now with our production department and you will be notified of the publication date in due course.

With kind regards,

Timea Kemeri-Szekernyes

PLOS Genetics

On behalf of:
